# Interdisciplinary Research to Advance Digital Imagery and Natural Compounds for Eco-Cleaning and for Preserving Textile Cultural Heritage

**DOI:** 10.3390/s22124442

**Published:** 2022-06-12

**Authors:** Dorina Camelia Ilies, Zlatin Zlatev, Alexandru Ilies, Berdenov Zharas, Emilia Pantea, Nicolaie Hodor, Liliana Indrie, Alexandru Turza, Hamid R. Taghiyari, Tudor Caciora, Monica Costea, Bahodiron Safarov, Barbu-Tudoran Lucian

**Affiliations:** 1Department of Geography, Tourism and Territorial Planning, Faculty of Geography, Tourism and Sport, University of Oradea, 1 Universitatii Street, 410087 Oradea, Romania; dilies@uoradea.ro (D.C.I.); ilies@uoradea.ro (A.I.); 2Faculty of Technics and Technologies, Trakia University, Bulgaria, 38 Graf Ignatiev Street, 8602 Yambol, Bulgaria; zlatin.zlatev@trakia-uni.bg; 3Department of Physical and Economic Geography, Faculty of Science, L.N. Gumilyov Eurasian National University, 2 Satpayev Street, Nur-Sultan 010008, Kazakhstan; berdenov-z@mail.ru; 4Faculty of Environmental Protection, University of Oradea, Magheru Street 26, 410087 Oradea, Romania; emilia.pantea@uoradea.ro (E.P.); mcostea@uoradea.ro (M.C.); 5Faculty of Geography, Babes-Bolyai University, 5-6 Clinicilor Street, 400006 Cluj-Napoca, Romania; nicolaie.hodor@ubbcluj.ro; 6Department of Textile, Leather and Industrial Management, Faculty of Energy Engineering and Industrial Management, University of Oradea, B. St. Delavrancea Str. No. 4, 410058 Oradea, Romania; lindrie@uoradea.ro; 7National Institute for R&D of Isotopic and Molecular Technologies, 67-103 Donat Street, 400293 Cluj-Napoca, Romania; alexandru.turza@itim-cj.ro (A.T.); lucian.barbu@itim-cj.ro (B.-T.L.); 8Faculty of Materials Engineering & New Technologies, Shahid Rajaee Teacher Training University, Tehran 16788-15811, Iran; htaghiyari@sru.ac.ir; 9Department of Digital Economy, Samarkand State University, Samarkand 140104, Uzbekistan; safarovb@rambler.ru; 10Electron Microscopy Center “Prof. C. Craciun”, Faculty of Biology & Geology, “Babes-Bolyai” University, 5-7 Clinicilor Str., 400006 Cluj-Napoca, Romania

**Keywords:** heritage textiles, digital imagery, SEM, artificial intelligence techniques, X-ray powder diffraction technique

## Abstract

The old fibers that make up heritage textiles displayed in museums are degraded by the aging process, environmental conditions (microclimates, particulate matter, pollutants, sunlight) and the action of microorganisms. In order to counteract these processes and keep the textile exhibits in good condition for as long as possible, both reactive and preventive interventions on them are necessary. Based on these ideas, the present study aims to test a natural and non-invasive method of cleaning historic textiles, which includes the use of a natural substance with a known antifungal effect (being traditionally used in various rural communities)—lye. The design of the study was aimed at examining a traditional women’s shirt that is aged between 80–100 years, using artificial intelligence techniques for Scanning Electron Microscopy (SEM) imagery analysis and X-ray powder diffraction technique in order to achieve a complex and accurate investigation and monitoring of the object’s realities. The determinations were performed both before and after washing the material with lye. SEM microscopy investigations of the ecologically washed textile specimens showed that the number of microorganism colonies, as well as the amount of dust, decreased. It was also observed that the surface cellulose fibers lost their integrity, eventually being loosened on cellulose fibers of cotton threads. This could better visualize the presence of microfibrils that connect the cellulose fibers in cotton textiles. The results obtained could be of real value both for the restorers, the textile collections of the different museums, and for the researchers in the field of cultural heritage. By applying such a methodology, cotton tests can be effectively cleaned without compromising the integrity of the material.

## 1. Introduction

Fiber’s constituent of heritage textiles weathering can be caused by the environmental conditions (microclimate, particulate matters, pollutants, sunlight) and the aging process of the cotton fibers, resulting in a decline in their strength, elasticity and cohesion level [1,2,3]. Cotton fibers composed of natural organic material (90% cellulose) are also subjected to the degrading effects of microorganisms in storage or exhibition places [4,5]. The undesirable consequences can consist of discoloration, loss of fibers luster, modification of tensile properties and weight modifications. Textural features can determine changes in textile fabrics after various treatments. They can be linked to the technical measurements of the characteristics of textile fabrics to obtain detailed information about the changes in these characteristics [6,7]. Digitization can be a significant apparatus in the present endeavors toward the promotion, preservation and renovation of cultural heritage [8,9,10,11]. By digitizing the heritage textiles and making the information accessible to all interested parties, they help to protect our cultural heritage. The study of cultural heritage artifacts and the preparation of conservation and restoration interventions are often limited to surface characterization of the constituent material of which the object is made. The value of the physico-chemical investigations at the surface level of the artifacts brings out important information. This is about their physical constitution, authenticity, history, elaboration and behavior after they have been abandoned and/or stored; its usefulness is to increase the knowledge about civilization, history and evolution, execution techniques and for the development of an optimal conservation policy [1,3].

In order to obtain information and imaging for the digitization of heritage objects, in recent years, good results have been obtained through photography and video, as well as through analysis and interpretations obtained, especially through the application of modern non-destructive techniques, such as, X-ray techniques applications, infrared radiation (compound crystallography, ≥10 µm), optical microscopy, thermal analyses, transmission electron microscopy (TEM), reflectance FT-IR (compound and bonds, analyzed depth µm level), (TEM), atomic force microscopy (AFM) (investigations at the level of the first layer of surface topography), nuclear magnetic resonance (NMR) method and laser 3D scanning [12,13,14]. Asmus and Katz [15], Comelli et al. [16], Filippidis et al. [17] and Aucouturier and Darque-Ceretti [18] used digital image processing and physico-chemical investigations to determine the degree of degradation and the measures needed to be taken on the surfaces of works of art, especially valuable paintings. Regarding the use of three-dimensional models (photogrammetry and 3D modeling) applied to cultural heritage objects, Manferdini and Remondino [19], Jiménez Fernández-Palacios et al. [20] and Herman et al. [21] use them for degradation assessment, virtual promotion and long-term conservation applications, while Higueras et al. [22] and Albu et al. [23] use them for the remote restoration of movable and immovable cultural heritage. Ortiz et al. [24], Peets et al. [25], Demenchuk et al. [26] and Zhou et al. [27] seek, in their infrared spectroscopy studies, to evaluate and identify the composition of old textiles and paintings, some of which are made of canvas. Other studies focus on the separate or combined use of SEM and X-ray in the field of cultural heritage in general and in the case of historical textiles in particular [28,29,30,31].

The main application of scanning electron microscopy (SEM) for fibers was in the medium to large magnification range and with greater depths of focus than optical microscopy. The resolution has been much improved, and the sensitivity increased, which leads to a considerable limitation of the exposure times of the samples and implicitly of the least possible damages that can be caused by radiation [14,32]. In order to carry out high-precision studies, the samples must be very thin (thickness less than 0.1 µm) and of certain dimensions to allow them to be fixed in the special device (from this point of view the technique is considered micro-invasive) [14], for facilitating the transmission of electrons, as well as for more precision, with a great depth of focus obtaining information about the morphology of the scanned areas. The problem facing electron microscopy for the study of textile fibers is securing a sufficiently good contrast; this can be achieved either by the technique of coloring with heavy metal compounds or the technique by which the effect of shadows is created by depositing the atoms of heavy metals on the specimen at certain angles. Another possibility would be to apply the technique, which consists of analyzing the remaining fragments as a result of the bio-chemical-mechanical degradation of the fibers. The polymers in the fiber composition can also be investigated as very thin films. SEM allows focusing an image, resulting in the electron diffraction pattern of the crystal lattice, orientation and crystallinity information even from a certain area of an image (similar to those resulting from the application of the X-ray diffraction technique), not necessarily from the whole specimen [32], and information on the chemical composition of inorganic compounds [14].

Most studies in the literature combine digital imaging techniques (X-ray Spectroscopy, SEM, Fourier transform infrared spectroscopy, etc.) with non-invasive natural substances for the treatment and preservation of heritage textiles. Table 1 presents, in detail, some of the works that use modern digital techniques to investigate the state of conservation of textiles and non-destructive solutions for the dual objective of cleaning–preservation.

As noted above, this study also combines the digital techniques of analysis and interpretation (SEM and X-rays) of the state of conservation of textile materials belonging to the Romanian cultural heritage with the traditional methods used in the past to clean these garments. Lye is the traditional cleaning method used as it is known as a natural substance with antifungal properties.

The main purpose of the study is to observe to what extent these traditional methods and substances for cleaning old materials are effective or not. SEM analyses were performed both before washing the material with lye and after this intervention to compare the results and establish the cleaning and antifungal properties of the natural substance. At the same time, the data obtained were analyzed by various imaging and statistical methods to identify the extent to which the applied treatment directly degrades the constituent cotton fibers, thus having destructive potential on the entire garment.

The ultimate goal of the research is to identify new and non-destructive methods of cleaning and long-term preservation of historical textiles in museums. In this context, most textile exhibits are treated with various cleaning chemicals, most of which have strong side effects that can eventually lead to the destruction of the constituent material.

## 2. Materials and Methods

The heritage textile object under investigation is from the Bihor region, Romania (Figure 1). It is a female cotton shirt, woven in a house, aged 80–100 years old and stored in an ethnographic museum. The fabric was investigated before and after natural, ecological washing with “lye”.

Lye is an ecological substance obtained from boiled ash. The process is to boil the ash with 1 L to 10 L of clear water from a river or a well. The lye results from the burning of trunks (saw-cut logs and axe splits) and thick-cut beech branches (Fagus sylvatica), with the whole well-dried bark, in traditional household stone or brick stoves. For bleaching the cloth, this solution was traditionally used in different parts of Romania, both immediately after their sewing and during the long period of use. It is important that the thick toile is not washed with lye and only the cloth, the shirts, the traditional men’s trousers (wide summer trousers) and the “bed sheet”. It is usual to soak the shirt/coat for 30 min in lukewarm water and then insert it into a “barrel” made of narrow wooden planks, slightly rounded in width (like a circular arch) or finished obliquely at the joints to form a round vessel. The bottom of the barrel has holes in it so that all the lye drains well.

The lye solution is poured from above, through a homemade gauze, over the shirt. This action is usually performed in the evening. In the morning, the coat is removed from the “barrel”. Therefore, the piece of clothing stays in the lye and in the lye steam for about 8 or 10 h. Hot water/hot lye negatively affects the colored parts of the garment/shirt, which is why the warm water solution is applied. Most of the washing and cleaning process was performed in river water.

The shirt is tightened by twisting and hitting with a piece of beech wood (a rectangular bat with carved and engraved ornaments, placed on the sides that do not touch the canvas), very often, on a thick slab of andesite stone, eroded by the river, carefully chosen and placed at a 45-degree inclination to the horizontal plane or hit on a long chair made of thick beech board, specifically made for such uses. Thus, the water is released with each hit. The operation is performed by alternately hitting, rinsing, rubbing, squeezing and soaking as many times as needed until cold clear water flows out of it.

Each piece should be washed separately and not mixed with clothes of different colors. The garments should never be put in a washing machine, as certain parts that adorn the clothes have holes, lace, different decorative handmade decorative models, and sometimes beads, which could be damaged by the mechanical action during operation and spinning in the washing machine. Less often, if necessary, it is finally washed with homemade soap made from animal bones by boiling it in rainwater. The shirt is dried in the sun, hung on a hanger or on a wooden beam so that the weight of the water straightens it and minimal, if any, ironing is needed.

The morphology and qualitative analysis of fibers were carried out using a scanning electron microscope (SEM) Hitachi SU8230 cold field emission gun STEM (Chiyoda, Tokyo, Japan). The samples were sputter-coated with 6 nm gold (Agar Automatic Sputter-Coater, Stansted, Essex, UK) for better conductivity required for high-resolution SEM imaging. SEM analysis parameters were inter alia, high vacuum mode, 30 kV acceleration voltage, secondary electron detectors (upper and lower) and two magnification orders, one for general aspects of samples and a higher one for surface topography analysis.

Table 2 describes the steps used in data processing from SEM images of washed and unwashed textiles. Data processing involves the use of global texture features from which feature vectors are formed. The data from them were reduced by the PCA method; the kernel variant was used. The reduced data are classified. An appropriate classification algorithm has been selected, and recommendations for practice have been made.

Kernel variant of principal component analysis (kPCA) [39]. One of the widely used methods of training based on templates is the kernel analysis of the principal components. It is an extension of the PCA using kernel techniques. Using a single kernel, the original PCA transformation is performed in a new high-dimensional space, with nonlinear mapping of the input data.

PCA starts with the calculation of the covariance matrix *C* of the matrix *X* of the input data with dimension *m × n*:(1)$$C=\frac{1}{m}\overset{m}{\underset{i=1}{\sum }}x_{i}{x_{i}}^{T}$$

The data are then projected onto the first k eigenvectors of the matrix. kPCA starts by calculating the covariance matrix *C* of the data after they have been transformed into a high-dimensional space *Φ*(*x*)*:*(2)$$C=\frac{1}{m}\overset{m}{\underset{i=1}{\sum }}\Phi \left(x_{i}\right)\Phi {\left(x_{i}\right)}^{T}$$

The transformed data are then projected onto the first k eigenvectors of the matrix, similar to PCA. The kernel trick method is used to decompose some of the calculations so that the whole process is implemented without actually calculating *Φ*(*x*)*. Φ* must be selected as a known kernel. In this work, Simple, Polynomial and Gaussian kernel functions were used.

Naïve Bayesian classifier (NB) [40]. Based on Bayes’ theorem for determining the a posteriori probability of an event, this classifier has become one of the classic algorithms in machine learning. By accepting the “naive” assumption of conditional independence between each pair of attributes, the naive Bayesian classifier effectively handles too many attributes to describe an example, i.e., with the so-called “Curse of dimension”. Bayes’ theorem:(3)$$P\left(y=c|x\right)=\frac{P\left(x|y=c\right)P\left(y=c\right)}{P\left(x\right)}$$
where *P* (*y* = *c*|*x*) is the probability that the object belongs to class *c* (a posteriori probability of the class); *P* (*x*|*y* = *c*)—the probability that the object *x* will meet in the middle of the object of class *c*; *P* (*y* = *c*)—unconditional probability to find object *y* in class *c* (a priori probability of the class); *P* (*x*)—unconditional probability of the object *x*.

The purpose of the classification is to determine to which class object *x* belongs. Therefore, it is necessary to find the probability class of the object *x*, i.e., it is necessary to choose from all classes the one that gives the maximum probability *P* (*y* = *c*|*x*).
(4)$$c_{opt}=\begin{array}{c} argmax\\ c\in C \end{array}\;P\left(x|y=c\right)P\left(y=c\right)$$

Discriminant analysis (DA). The definition of discriminant functions is performed by discriminant analysis using linear (LDA) and quadratic (QDA) separation functions [41].

QDA is a better option than LDA for large data sets. This is because the QDA tends to have a lower deviation and higher variation. On the other hand, the LDA is more suitable for smaller data sets that have higher grouping and lower variance. In summary, the quadratic separating function has the form:(5)$$\delta_{k}\left(x\right)=-\frac{1}{2}{\left(x-\mu_{k}\right)}^{T}{\Sigma_{k}}^{-1}\left(x-\mu_{k}\right)+\log\left(\pi_{k}\right)$$
where *δk* is a separating function; *µk*—averaged vector; *x*—observations; *Σ^−1^*—covariance matrix.

For practical purposes, including the creation of control programs for driver assistance systems, it is convenient to present the separation function in the form of:(6)$$\delta \left(x\right)=K+v\cdot L+v^{\prime }\cdot Q\cdot v)$$
where *K* is a constant; *L*—linear coefficient; *Q*—square coefficient; *v* = [*x;y*]—vectors (matrix) of data; *x* and *y* are the data along both axes; *v’*—transposed matrix of *v*.

Kernel SVM (KSVM) [42]. Using the core support vector machines method (KSVM), nonlinear transformation of the original data into another higher-dimensional space is performed, where the objects are linearly separable. In SVM, hypersurfaces separating the classes for which the distance between the boundaries for both classes are maximal can be calculated from reference points that represent boundary points for a given data class in the multidimensional feature space.

According to the type of the selected kernel function *Φ*, several types of classifiers are constructed (linear, radial-basis function, polynomial, neural network). A width σ is chosen, and a kernel function with the following general form:(7)$$K\left(x,y\right)={\left(x^{T}y+C\right)}^{d}$$
where *K* is a kernel function; *x* and *y* are input vectors (vectors of features determined by the training sample); *C* > 0 is a constant. At *C* = 0 the kernel is homogeneous.

Evaluation of classifiers [42]. Basic (*e_i_*), actual (*g_i_*) and total (*e*_0_) classification errors were used to evaluate the performance of the classifiers. The input data for the two classes processed by the classifier can be in groups: correctly (Positive *P*) and incorrectly (Negative *N*) classified.

The errors can be defined by:(8)$$e_{i}=\frac{FN_{i}}{TP_{i}+FN_{i}}\times 100,\%$$
(9)$$g_{i}=\frac{FP_{i}}{TP_{i}+FP_{i}}\times 100,\%$$
(10)$$e_{0}=\frac{\sum_{i=1}^{m}FN_{i}}{\sum_{i=1}^{m}TP_{i}+\sum_{i=1}^{m}FN_{i}}\times 100,\%$$

The X-ray diffraction patterns scan was achieved in reflection Bragg–Brentano geometry; the samples are fixed in the sample holder such that the surface of the analyzed specimen is attached parallel to the support. The measurement of X-ray powder diffraction patterns was performed with a Bruker D8 Advance diffractometer equipped with a LINXEYE detector, a germanium (1 1 1) monochromator and an X-ray tube operating at 40 kV and 40 mA. The acquisition was performed with CuKα1 radiation in DIFFRAC plus XRD Commander Program considering a scan rate of 0.01°/s.

Methods of consistently improving assessments have been used for the selection of informative textural features. They significantly reduce the number of resulting combinations of traits used for classification. The FSNCA, RelieFf and SFCPP methods described in detail in [43] were used. Method for selection of features by analysis of adjacent components FSNCA. This algorithm is suitable for assessing the significance of characteristics of distance-based models. Method for ranking significant parameters for RelieFf forecasting. The method is a selection function using RelieFf’s classification algorithm. It is suitable for assessing the significance of features for distance-based models. Method of the subsample of traits with comparable predictive power SFCPP.

## 3. Results and Discussions

SEM provides useful tools for examining the surface and structural characteristics of fibers of the fabrics through high resolution and depth of field of images. The scanning electron micrograph (Figure 2a) of convolutions in mature cotton fiber, bar = 18.9 µm, shows that cotton fibers look like collapsed and twisted tubes (blue arrow). The longitudinal view of organic cotton fibers shows ribbon-like twists (lower part of the image-orange arrow). The deterioration of the fibers (longitudinal)—detail of fibers—and the peeling process (black arrow, more visible in the upper left part of this image) is evident [44]. In Figure 2b, staple fiber ends of cotton fibers damaged in cotton fibers are damaged, shown with black lines on the figure (black arrow) [45]. The deterioration of the fibers (longitudinal), detail of fibers and the peeling process (orange arrow) are shown, as are dust/microorganisms on the fibers. Some fibers (Figure 2c,d) appear like collapsed and twisted tubes. These seem to have deteriorated and the peeling process is very intense (black arrow); note the presence of a high quantity of dust/microorganisms (fungi upper and lower right part of the image). Organic cotton fibers showing ribbon-like twists (left part of the image) as well as dust and/or microorganisms present on the cotton fibers are observable. In addition, the peeling process of the fibers is visible (Figure 2e).

Staple fiber ends–end of cotton fibers (Figure 3a) are damaged in processing; organic cotton fibers look like collapsed and twisted tube or show ribbon-like twists. Less evidence of dust/microorganisms on the fibers after the wash [33] (Figure 3b). Of note is the peeling of the fibers and possible dust presence on the fibers in the upper part of the image. It can also be seen showing cross-marking or nodes. Longitudinal cotton fibers (Figure 3c–e) appear undeteriorated and ”more clean” than the previous images of unwashed specimens showing peeling of the fibers and the limited existence of dust/microorganisms. The appearance of cotton fibers (Figure 3f) under the electronic microscope show longitudinal cotton fibers are present and in good condition. Dust/microorganisms can be identified in the central part of the image, with the peeling process of the fibers in evidence on the lateral margins.

The comparison between the washed and unwashed images demonstrates that some cellulose fibers and microfibrils have lost their integrity with the main cotton fibers, being loosened and eventually separated from the cotton fibers. This separation of cellulose microfibrils is apparent in the comparison made in Figure 4a,b.

Considering the fact that cellulose fibers and microfibrils are mainly connected to each other in a cotton fiber by hydroxyl bonds, ultimately forming the final cotton fiber, the process of washing the cotton textile specimens in water (with abundant free hydroxyl groups –OH) facilitated this easy loosening and the eventual separation of the surface microfibrils. Similar breakage of primary bonds between wollastonite and wood polymers (cellulose and hemicellulose) was previously reported as a result of abundant water molecules, though the adsorption energy of wollastonite–cellulose was calculated to be greater than that of water–cellulose (red arrows) [46,47,48]. The comparison of the SEM image of washed cotton textile specimen (Figure 5a) with that of an unwashed specimen (Figure 5b) with 50-micron magnification clearly illustrates that no separation of cellulose microfibrils can be observed in the unwashed specimen. As an example, some defects on the fibers are shown with red lines.

It is important to note that the loosening and separation of surface cellulose microfibrils cannot solely be attributed to the abundance of water hydroxyl groups, the breakage of hydroxyl bonds that made cellulose fibers and microfibrils integrated to form cotton fibers. The friction between cotton fibers during washing can be considered another cause for the breakage and the eventual separation of microfibrils.

Analysis of SEM images of historical textiles using artificial intelligence techniques. SEM images of washed and unwashed textile fabrics were used. The photos were in TIFF file format. They had resolutions of 1200 × 960 pix. They were divided into three groups depending on the magnification ×1000, ×300 and ×100 LM. In the analysis of the SEM images of textiles, the most commonly used way of their digital representation is textural features [37,38]; the advantage of using textural features in the analysis of SEM images is that they can be used to analyze, model and process the texture. In this way, human vision is simulated by distinguishing elements in images. Texture features provide sufficient information about objects in SEM images appropriate for their classification, identification and prediction modeling.

Twenty-two texture features were used, described in detail [49]. These are described in general terms with their formulas (Table 1: Equations (1)–(22)), where *μx*, *μy*, *σx*, *σy* represent the mean values and standard deviations of px and py functions of the partial probability density; *x* and *y* are coordinates (rows and columns) of a common matrix; probability “*p*”, *px* + *y*(*i*) is the probability of the joint matrix; HX and HY—entropy *of px* and *py*; *N* is the number of grey levels in the image.

Table 3 shows the mean and standard deviation of the two classes of textile fabrics—unwashed (UW) and washed (W). Some of the features have similar values, while others visibly overlap. It is necessary to make a selection of these textural features that are sufficiently informative and can be used to classify both types of fabrics. The names of texture features are according to Boland [49]. All names are in their original form.

It is an optimal set of features that are mutually and maximally different and can effectively represent the compared objects through it (Table 4).

Table 5, Table 6 and Table 7 show the results of a selection of informative textural features, depending on the resolution of the SEM images and the magnification at which they were obtained. The smallest number of features (five) were obtained by the FSNCA method. It is followed by the RelieFf method, with seven features in all cases. The most features were selected by the SFCPP method, a total of 13 features, regardless of image resolution.

The kernel variant of the principal component (kPC) method was used to reduce the data volume of the resulting feature vectors [40]. When reducing data from vectors with textural features, the use of the method is appropriate because the data have relatively complex structures that cannot be represented with sufficient accuracy in a linear subspace. The kernel functions used are three linear (Simple) and two nonlinear: Polynomial (Poly) and Gaussian. These functions correspond to the way of design (transformation) in the space of the principal components.

Figure 6 shows the reduced data from a vector with textural features selected by the FSNCA method for three principal components with a linear (Simple) kernel. The visualization depends on the magnification of the images. There is a visible separation between the two types of textile fabrics observed. Classification methods will highlight this distinctiveness. The scattered data in Figure 6 obtained from the washed and unwashed specimens reveal distinct separation between the data grouping of the unwashed specimens for all three principal components with a linear kernel (×100, ×500 and ×1000 LM). For the washed data, though, the data grouping revealed a less degree of distinction; that is, data are scattered more widely in comparison to the data of the unwashed specimens. The widely scattered data for the washed specimens clearly indicate that the breakage of hydroxyl bonds has happened on a random basis along cellulose microfibrils along cotton threads, which is common in natural materials [46,47,48]. Still, it can be observed that no overlap can be found between the two main data groupings of the washed and the unwashed observations. From a statistical point of view, the data in each group of the washed and unwashed can be considered sufficient to come to a conclusion that there is a statistically significant difference between them from all three principal components with a linear kernel. It is to be noted that the physical breakage of microfibrils by friction has intensified the wide scattering of data for principal components with a linear kernel in the washed specimens.

The methods of naïve Bayesian classifier, discriminant analysis and support vector machines method (SVM) were used for classification [40]. Two types of separating functions were used in the discriminant analyses—linear and quadratic. The SVM method uses three separating functions—linear, quadratic and one based on radial basis functions.

Figure 7 shows an example of data classification for washed and unwashed textile fabrics obtained from textural features of their SEM images. They have a magnification of ×1000 LM. The vector of features was selected by the RelieFf method. It is reduced to principal components with a linear (Simple) kernel. Linear separation functions were used in the discriminant analysis and the support vector machines method. The Bayesian classifier shows the restriction of the areas of the two classes of textile fabrics with spherical borders. With regard to the covariance nature of the principal component, the absolute values for PC1 and PC2 may range from 0 to 1. In the case of unwashed specimens, the values were very near to 0, indicating that there was little or no covariance between PC1 and PC2. However, the scattered dots for the PC1 and PC2 values for the washed specimens meant a quite different trend. This clearly illustrated a significant difference between the washed and the unwashed specimens (Figure 7b).

An assessment of classification errors has been made [42]. These errors are basic (*e_i_*), actual (*g_i_*) and general (*e*_0_). When calculating these errors, the false-negative (FN) data, which are the number of Class *i* data incorrectly assigned to other classes, as well as the true-positive (TP) data, which are the number of correctly classified Class *i* data, are determined.

The results of a naive Bayesian classification are shown in Table 8. Using a linear kernel of the main components between the two groups of tissues results in zero classification errors. The same trend is observed when using reduced data from the vector of features with principal components using a Polynomial kernel. The highest values of classification errors are obtained when using data reduced by main components with Gaussian kernel. The actual error of over 40% obtained between all the surveyed areas shows that a large part of the data from the second class fell into the first. Hence, the total classification error reaches values of 27–52%, which is an indication that the reduction of data with this method is inappropriate.

The results of the discriminant analysis classification are shown in Table 9. The total classification error is 18–24%. Using a feature vector selected by the RelieFf method and reduced with principal components using a Polynomial kernel, maximum values of the total classification error of 24% are obtained. The basic error, in this case, reaches 8%. This shows that 8% of the first class of textile fabrics data falls into the second. Classification errors above 10% indicate that using this classifier to separate the two studied classes of textile fabrics is not appropriate.

The classification results with the support vector machines method are presented in Table 10. In the general case, zero classification errors occur. Error values above 10% are obtained using a linear separator function of the classifier. In this case, the actual error is 12–29%, which indicates that when using the linear separation function of the SVM classifier, some of the data is incorrectly classified and falls into the class to which they do not belong.

The obtained results found that the separation of the two classes of textile fabrics—washed and unwashed, does not depend on the method of selecting informative textural features. It also does not depend directly on the classifier used. Suitable methods for reducing the data volume of feature vectors are principal components with linear and Polynomial kernels. It is not appropriate to use linear separation functions of classifiers because high values (above 10%) of classification errors are obtained.

The X-ray powder diffraction technique was used for the structural characterization of heritage cotton textiles. In the case of textiles, they contain both crystalline and amorphous phases. The crystalline phase is highlighted by diffraction peaks corresponding to the crystallographic planes involved and the amorphous phase produces diffraction halos. X-ray diffraction is specific for each class of textile materials, thus characterizing these materials in terms of crystalline peaks and amorphous halos; a structural fingerprint of these materials is obtained. This paper characterizes the crystalline phase and the amorphous phase of materials, evaluating the degree of crystallinity and the dimensions of crystallites corresponding to the crystalline phase at the same time.

The X-ray diffraction patterns for specimens 1 and 2, both before and after washing, are shown in Figure 8a,b. In both cases, the existence of broad diffraction peaks specific to cellulose is observed [50]. According to the literature, cellulose is found in the monoclinic system, space group P21 with the following lattice parameters a = 7.784 Å, b = 8.201 Å, c = 10.380 Å, γ = 96.5° [51]. In this sense, the observed intense diffraction lines possess the following 2*θ* angular positions and have been assigned the following Miller indices: 15.1° (1 −1 0), 16.8° (1 1 0), 20.85° (1 0 2), 22.98° (2 0 0), 34.8° (0 0 4). The additional diffraction line at 2*θ* = 12.35° for the black sample may be due to the black pigment in the seam.

The crystallite sizes were evaluated using the Scherrer relation [52] (Equation (1)) for the (2 0 0) diffraction peak occurring at 2*θ* = 22.80° (where *λ* is defined as the wavelength of X-ray radiation, *β* is the full width at half maximum of the selected peak and *θ* represents the diffraction angle). The following results were obtained: D = 73 Å for the white sample and D = 45 Å for the black one, both before and after washing.
(33)$$d=\frac{0.9\lambda }{\beta cos\theta }$$

The crystallinity was evaluated based on Equation (2), where Icryst is the sum of the intensities due to the crystalline phases and Itotal is the sum of the intensities due to the crystalline phases plus the diffraction intensity due to the amorphous halos. A value of 34% was obtained for the white specimen and 29% for the black one, both before and after washing. The figure represents the X-ray diffraction patterns of investigated samples. The presented results are according to the color of the samples.
(34)$$Index=\frac{I_{cryst}}{I_{total}}\times 100\%$$

## 4. Conclusions

Following the proposed experiment, the following conclusions can be drawn:From SEM images, the cotton fibers are in the form of collapsed, ribbon-like twists and twisted tubes and no significant surface modifications were noticed after fiber washing.In the unwashed specimen, a fairly large number of microorganism (fungi) colonies, as well as a large amount of dust, were highlighted.After washing, the SEM microscopy showed that the number of microorganism colonies, as well as the amount of dust, decreased, the cellulose fibers lost their integrity and the presence of microfibrils that connect the cellulose fibers is better observed.X-ray diffraction showed that the basic material of the textile samples is cellulose, which possesses both a crystalline and an amorphous phase.The calculation of the degrees of crystallinity in the washed and the unwashed specimens revealed that the washing process did not significantly alter the degree of crystallinity.The crystallite dimensions were evaluated as being 73 Å for the white specimen and 45 Å for the black one, both before and after washing.Comparing the transverse dimensions of the cellulose fibers in the SEM images, which are of the order of 100–200 μm, it is found such a fiber consists of a very large number of crystallites.

Overall, the separation of the two classes of textile fabrics, washed and unwashed, does not depend on the method of selecting informative textural features, which are proposed by artificial intelligence methods. It also does not depend directly on the classifier used. Appropriate methods for reducing the data volume of selected texture feature vectors have been shown to be principal components with linear and Polynomial kernel functions. Reducing the data in the selected feature vectors by Gaussian kernel principal components is impractical. When using this method, classification errors of more than 10% occur. The use of the LDA classification method leads to classification errors above 10%, which makes its use inappropriate for the task solved in the present work. The reason for this could be that LDA is sensitive to overlapping data, missing values and deviations. The overlap of data is observed in the texture characteristics of the studied washed and unwashed textile fabrics. The lowest values of classification errors were obtained using the SVM method, compared to the other two methods, LDA and QDA.

## Figures and Tables

**Figure 1 sensors-22-04442-f001:**
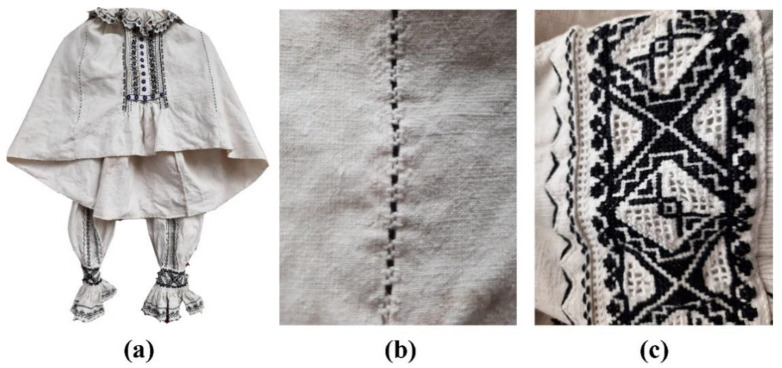
Female cotton shirt, woven in a house, aged 80–100 years old, from the Bihor region, Romania ((**a**)—overview of the appearance of the shirt; (**b**)—cotton sample; (**c**)—motifs sewn with black thread).

**Figure 2 sensors-22-04442-f002:**
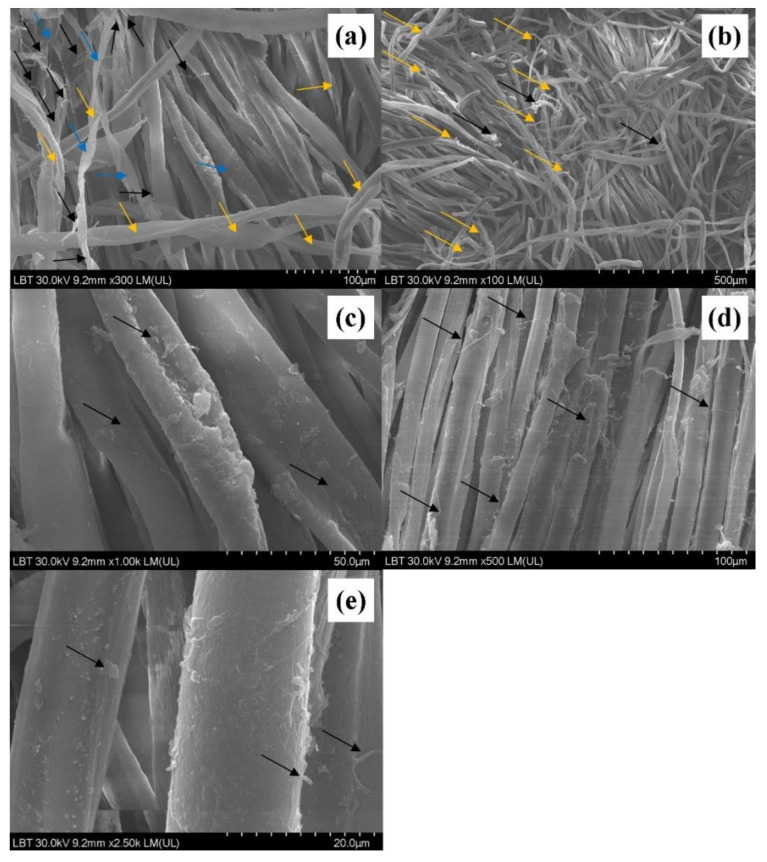
Scanning electron micrograph on the unwashed surfaces of the traditional shirt ((**a**)—magnitude ×300 LM; (**b**)—×100 LM; (**c**)—1.00K LM; (**d**)—×500 LM; (**e**)—×2.50 LM).

**Figure 3 sensors-22-04442-f003:**
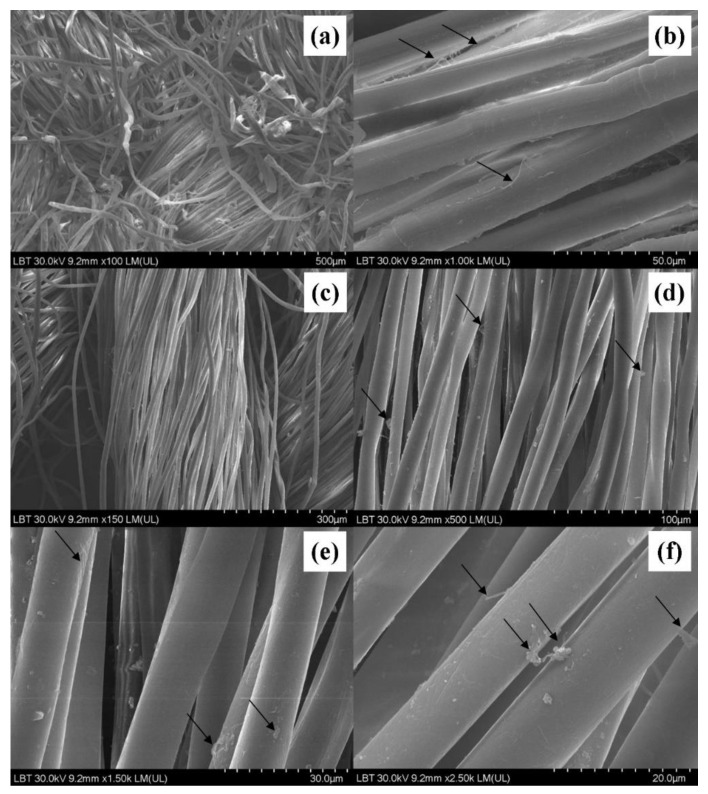
Scanning electron micrograph on the washed surfaces of the traditional shirt ((**a**)—magnitude ×100 LM; (**b**)—×1.00K LM; (**c**)—150 LM; (**d**)—×500 LM; (**e**)—×1.50K LM; (**f**)—×2.50K LM).

**Figure 4 sensors-22-04442-f004:**
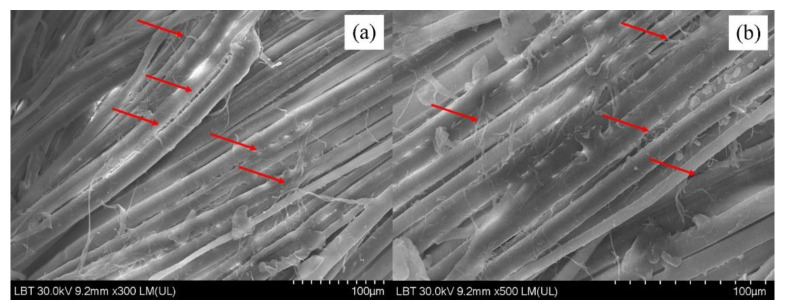
Comparison between unwashed and washed surfaces of the analyzed traditional shirt ((**a**)—unwashed specimen at ×300 LM magnitude; (**b**)—washed specimen at ×500 LM magnitude).

**Figure 5 sensors-22-04442-f005:**
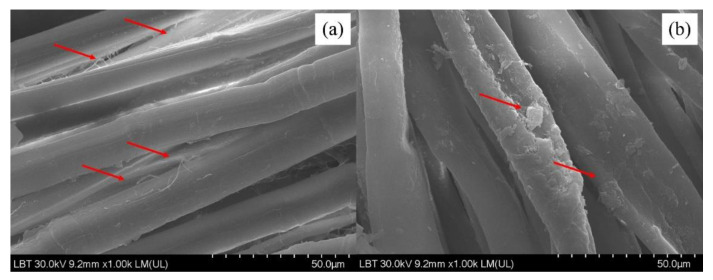
Comparison between unwashed and washed surfaces of the analyzed traditional shirt ((**a**)—washed specimen at ×1.0K LM magnitude; (**b**)—unwashed specimen at ×1.0K LM magnitude).

**Figure 6 sensors-22-04442-f006:**
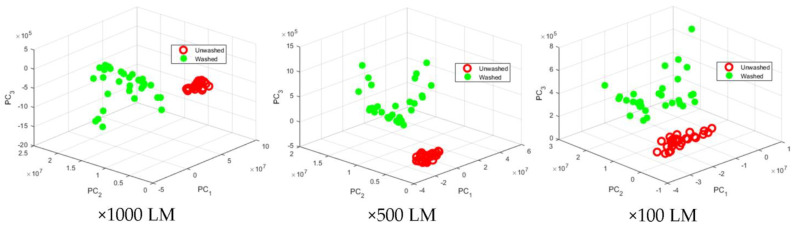
Presentation of reduced vector data with textural features on three principal components.

**Figure 7 sensors-22-04442-f007:**
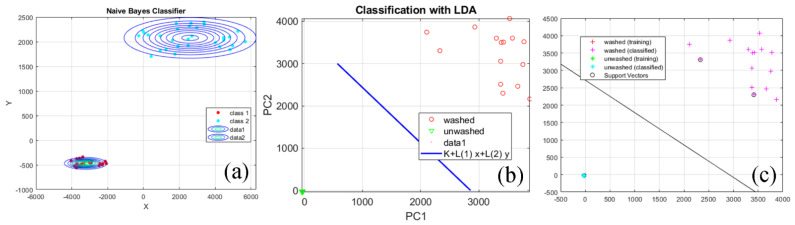
Example of classification of reduced data from a feature vector selected by a RelieFf method ((**a**)—Naïve Bayesian; (**b**)—Discriminant analysis; (**c**)—Support vector machines).

**Figure 8 sensors-22-04442-f008:**
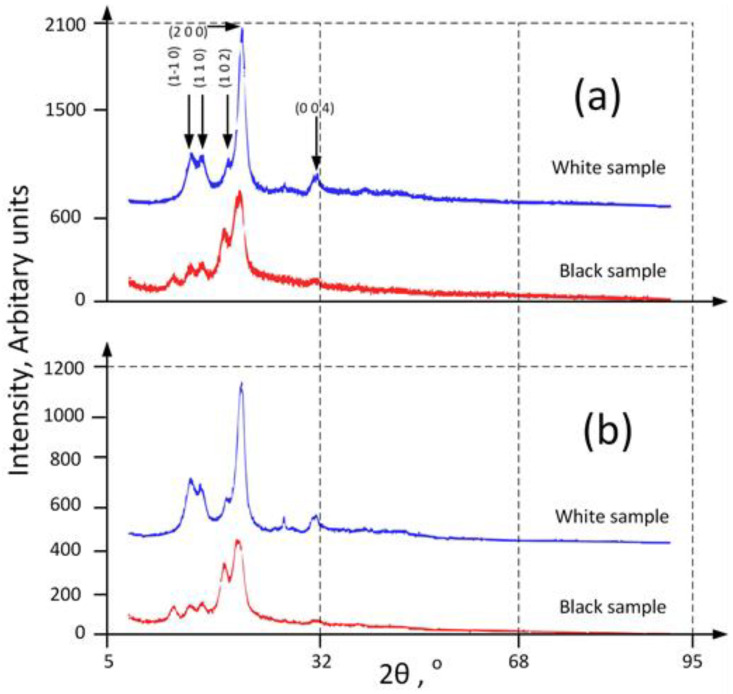
X-ray diffraction patterns of investigated samples ((**a**)—before washing; (**b**)—after washing).

**Table 1 sensors-22-04442-t001:** Presentation of the targeted textiles, methods and techniques used, purpose and findings of some important works in connection with the present study.

	Title, Year, Study Location	Targeted Samples	Methods and Techniques Used	The Purpose of the Research	Findings
**Investigating the state of conservation of textiles**	Comparative analysis of textile metal threads from liturgical vestments and folk costumes in Croatia, 2017, Croatia [33]	Textiles liturgical vestments	X-ray Spectroscopy, Rutherford Backscattering Spectroscopy	Obtaining valuable information about old manufacturing techniques.	The results are invaluable in selecting the right treatment for cleaning and preservation.
	Dust deposition on textile and its evolution in indoor cultural heritage, 2019, France [34]	Textiles stored in museums	SEM analysis	Investigating the degradation of heritage textiles due to the action of dust and chemical compounds.	The fibers themselves are not affected by gaseous pollutants, but the latter react with the particles of the dirty samples, leading to the formation of efflorescences.
	The application of FTIR microspectroscopy in a non-invasive and non-destructive way to the study and conservation of mineralised excavated textiles, 2019, Denmark [35]	Two textile samples excavated from old graves	Fourier Transform Infrared (FTIR) microspectroscopy	Investigating the degree of conservation of the targeted textile materials.	FTIR microspectroscopy applied in refectance mode is a non-invasive and non-destructive technique for analyzing fragile materials. Much of the organic matter in the fiber has been preserved at the molecular level, which would allow the safe application of any preservative treatments.
	Technical investigation and conservation of a tapestry textile from the Egyptian Textile Museum, Cairo, 2018, Egypt [36]	Textile tapestry dating from the 4–5th century AC	Scanning Electron Microscopy (SEM), EDAX (SEM–EDAX), Fourier Transform Infrared (FTIR)	Identification of textile fibers, damage to them and analysis of the mordant in the dyed samples.	SEM analyzes show that the fibers are very fragile and weak due to improper preservation. The FTIR results identify the brown source in the fabric as Indian cutch.
**Application of non-destructive solutions for textile cleaning and preservation**	Thyme essential oil for antimicrobial protection of natural textiles, 2013, Poland [37]	Various cotton fabrics	Scanning Electron Microscopy (SEM), EDAX (SEM–EDAX), Fourier Transform Infrared (FTIR), strength tests, antifungal and antibacterial tests, application of thyme essential oil.	Increasing the resistance of textiles to bacteria and mould action using natural compounds.	Thyme essential oil applied in a concentration of 8% in methanol, has shown antifungal and antibacterial effects among the strongest. The applied substance has inhibitory effects for certain types of molds, fungi and bacteria; effects that lead over time to the preservation of the support material.
	Investigations of the Surface of Heritage Objects and Green Bioremediation: Case Study of Artefacts from Maramures, Romania, 2021, Romania [3]	Romanian traditional sheepskin waistcoat aged 80–100 years old	Scanning Electron Microscopy (SEM) combined with application of six essential oils *(Lavandula angustifolia*, *Citrus limon*, *Mentha piperita*, *Marjoram*, *Melaleuca alternifolia*, *Origanum vulgare).*	Non-invasive cleaning of textile materials of microbiological flora in order to preserve the fibres.	The results show that these essential oils are an eco-friendly solution for cleaning historic textiles, being affordable and having very good antifungal and antibacterial effects, with effects that can last more than 30 days. At the same time, natural extracts have the potential to treat several different types of textiles.
	Antifungal activity of some plant extracts and essential oils against fungi-infested organic archaeological artefacts, 2019, Egypt [38]	Ancient papyrus and linen from Egyptian Museum, Cairo	Scanning Electron Microscopy (SEM) and Fourier transform infrared spectroscopy (FTIR) combined with application of nine kinds of powdered plant extracts and five essential oils.	Determination the antifungal effects of these substances against the most common fungi on heritage textiles (*Aspergillus flavus*, *A. versicolor*, *Penicillium* sp. and *P. purpurogenum*).	All applied substances have antifungal effects, but essential oils are shown to be very effective for the types of fungi identified. At the same time, the substances have a low toxicity and do not affect the support materials, while the risk of microorganisms developing resistance to them is quite low.

**Table 2 sensors-22-04442-t002:** SEM images data processing steps.

Stage	Description	Notes
Stage 1	Obtaining raw data for SEM images	Extraction of global texture features for washed and unwashed textiles
Stage 2	Selection of informative features	Through the methods of FSNCA, RelieFf and SFCPP
Stage 3	Creating three feature vectors	The features selected by the three methods form vectors of them
Stage 4	Reducing the volume of data in feature vectors	The kPCA method with Simple, Polynomial and Gaussian kernel functions were used
Stage 5	Selection of an appropriate method for reducing the volume of data in feature vectors	A reference naïve Bayesian classifier was used
Stage 6	Classification of washed and unwashed textile fabrics	Discriminant analysis and support vector machines methods were used
Stage 7	Choice of classification strategy	Justification for the choice of classifier and recommendations for practice

**Table 3 sensors-22-04442-t003:** Formulas of the used textures features (after Bolad [49]).

Autocorrelation	$T1=\underset{i}{\sum }\underset{j}{\sum }\left(ij\right)p\left(i,j\right)$	(11)	Maximum probability	$maxpr=\underset{i,j}{\max}\;p\left(i,j\right)$	(22)
Contrast	$T2=\sum_{n=0}^{N}n^{2}\left\{\sum_{i=1}^{N}\sum_{j=1}^{N}p\left(i,j\right)\right\},\left|i-j\right|$	(12)	Sum of squares: Variance	$T12=\underset{i}{\sum }\underset{j}{\sum }{\left(i-\mu \right)}^{2}p\left(i,j\right)$	(23)
Correlation 1	$T3=\underset{i,j}{\sum }\frac{\left(i-\mu_{i}\right)\left(j-\mu_{j}\right)p\left(i,j\right)}{\sigma_{i}\sigma_{j}}$	(13)	Sum average	$T13=\overset{2N_{g}}{\underset{i=2}{\sum }}ip_{x+y}\left(i\right)$	(24)
Correlation 2	$T4=\frac{\sum_{i}\sum_{j}\left(ij\right)p\left(i,j\right)-\mu_{x}\mu_{y}}{\sigma_{x}\sigma_{y}}$	(14)	Sum variance	$T14=\overset{2N_{g}}{\underset{i=2}{\sum }}{\left(i-cshad\right)}^{2}p_{x+y}\left(i\right)$	(25)
Cluster Prominence	$T5=\underset{i}{\sum }\underset{j}{\sum }{\left(i+j-\mu_{x}-\mu_{y}\right)}^{4}p\left(i,j\right)$	(15)	Sum entropy	$T15=-\overset{2N_{g}}{\underset{i=2}{\sum }}p_{x+y}\left(i\right)\log\left\{p_{x+y}\left(i\right)\right\}$	(26)
Cluster Shade	$T6=\underset{i}{\sum }\underset{j}{\sum }{\left(i+j-\mu_{x}-\mu_{y}\right)}^{3}p\left(i,j\right)$	(16)	Difference variance	$T16=\overset{N_{g}-1}{\underset{i=0}{\sum }}i^{2}p_{x-y}\left(i\right)$	(27)
Dissimilarity	$T7=\underset{i}{\sum }\underset{j}{\sum }\left|i-j\right|p\left(i,j\right)$	(17)	Difference entropy	$T17=-\overset{N_{g}-1}{\underset{i=0}{\sum }}p_{x-y}\left(i\right)\log\left\{p_{x-y}\left(i\right)\right\}$	(28)
Energy	$T8=\underset{i}{\sum }\underset{j}{\sum }p{\left(i,j\right)}^{2}$	(18)	Information measure of correlation 1	$T18=\frac{HXY-HXY_{1}}{\max\left\{HX,HY\right\}}$	(29)
Entropy	$T9=-\underset{i}{\sum }\underset{j}{\sum }p\left(i,j\right)\log\left(p\left(i,j\right)\right)$	(19)	Information measure of correlation 2	$T19=\sqrt{1-e^{-2\left(HXY2-HXY\right)}}$	(30)
Homogeneity 1	$T10=\underset{i,j}{\sum }\frac{p\left(i,j\right)}{1+\left|i-j\right|}$	(20)	Inverse difference normalized	$T20=\overset{N_{g}}{\underset{i=1}{\sum }}\overset{N_{g}}{\underset{j=1}{\sum }}\frac{p\left(i,j\right)}{1+\left(\frac{{\left|i-j\right|}^{2}}{{N_{g}}^{2}}\right)}$	(31)
Homogeneity 2	$T11=\underset{i}{\sum }\underset{j}{\sum }\frac{1}{1+{\left(i-j\right)}^{2}}p\left(i,j\right)$	(21)	Inverse difference moment normalized	$T21=\underset{i}{\sum }\underset{j}{\sum }p{\left(i,j\right)}^{2}$	(32)

**Table 4 sensors-22-04442-t004:** Mean value and standard deviation of textural features of SEM images of textile fabrics.

Magnification		×1000 LM		×500 LM		×100 LM	
	Type	UW	W	UW	W	UW	W
Feature							
*T*1		20.25 ± 3.03	20.22 ± 1.8	20.13 ± 2.84	20.28 ± 1.7	20.15 ± 2.74	20.29 ± 1.87
*T*2		0.39 ± 0.17	0.45 ± 0.19	0.48 ± 0.21	0.55 ± 0.24	0.7 ± 0.34	0.77 ± 0.28
*T*3		0.75 ± 0.1	0.75 ± 0.11	0.71 ± 0.15	0.7 ± 0.14	0.6 ± 0.19	0.58 ± 0.18
*T*4		0.74 ± 0.11	0.75 ± 0.12	0.7 ± 0.13	0.69 ± 0.15	0.59 ± 0.17	0.56 ± 0.18
*T*5		26.74 ± 3.06	37.68 ± 17.64	24.21 ± 2.76	32.46 ± 15.94	21.59 ± 2.76	30.11 ± 13.98
*T*6		3.74 ± 1.15	3.8 ± 1.91	3.37 ± 1.13	3.08 ± 2.26	2.95 ± 1.11	3.17 ± 1.76
*T*7		0.3 ± 0.1	0.34 ± 0.12	0.36 ± 0.13	0.4 ± 0.15	0.44 ± 0.19	0.5 ± 0.23
*T*8		0.27 ± 0.05	0.23 ± 0.05	0.26 ± 0.07	0.23 ± 0.05	0.23 ± 0.07	0.21 ± 0.07
*T*9		1.89 ± 0.18	2.04 ± 0.26	1.97 ± 0.24	2.1 ± 0.29	2.1 ± 0.33	2.24 ± 0.29
*T*10		0.88 ± 0.03	0.86 ± 0.05	0.86 ± 0.05	0.85 ± 0.08	0.81 ± 0.07	0.81 ± 0.07
*T*11		0.87 ± 0.04	0.86 ± 0.05	0.85 ± 0.05	0.84 ± 0.07	0.82 ± 0.07	0.8 ± 0.08
*T*12		0.45 ± 0.07	0.39 ± 0.07	0.43 ± 0.09	0.38 ± 0.08	0.4 ± 0.1	0.36 ± 0.08
*T*13		20.29 ± 2.92	20.46 ± 2.05	20.14 ± 2.82	20.64 ± 1.56	20.06 ± 2.63	20.48 ± 1.8
*T*14		8.87 ± 0.73	8.95 ± 0.55	8.85 ± 0.77	8.92 ± 0.37	8.84 ± 0.74	8.92 ± 0.51
*T*15		56.22 ± 9.7	54.93 ± 6.47	54.48 ± 8.74	54.61 ± 6.1	54.24 ± 9.2	54.88 ± 6.29
*T*16		1.59 ± 0.1	1.73 ± 0.11	1.62 ± 0.09	1.72 ± 0.14	1.64 ± 0.12	1.75 ± 0.16
*T*17		0.38 ± 0.17	0.44 ± 0.18	0.5 ± 0.25	0.55 ± 0.24	0.72 ± 0.35	0.76 ± 0.32
*T*18		0.68 ± 0.12	0.72 ± 0.14	0.72 ± 0.17	0.77 ± 0.16	0.85 ± 0.21	0.92 ± 0.2
*T*19		0.37 ± 0.11	0.34 ± 0.12	0.3 ± 0.13	0.28 ± 0.15	0.22 ± 0.13	0.22 ± 0.18
*T*20		0.75 ± 0.08	0.77 ± 0.1	0.72 ± 0.13	0.73 ± 0.17	0.62 ± 0.17	0.65 ± 0.23
*T*21		0.97 ± 0.01	0.97 ± 0.01	0.97 ± 0.01	0.96 ± 0.02	0.96 ± 0.02	0.95 ± 0.02
*T*22		0.99 ± 0	0.99 ± 0	0.99 ± 0	0.99 ± 0	0.99 ± 0.01	0.99 ± 0

**Table 5 sensors-22-04442-t005:** Selected texture features on images with magnification ×1000 LM.

Method for Selection	Selected Features
FSNCA	*T*5, *T*6, *T*9, *T*13, *T*15
RelieFf	*T*5, *T*6, *T*8, *T*9, *T*15, *T*16, *T*22
SFCPP	*T*2, *T*3, *T*4, *T*5, *T*8, *T*9, *T*10, *T*12, *T*13, *T*15, *T*17, *T*18, *T*21

**Table 6 sensors-22-04442-t006:** Selected texture features on images with magnification ×500 LM.

Method for Selection	Selected Features
FSNCA	*T*5, *T*6, *T*9, *T*16, *T*19
RelieFf	*T*5, *T*6, *T*8, *T*12, *T*15, *T*16, *T*22
SFCPP	*T*3, *T*4, *T*5, *T*8, *T*10, *T*11, *T*12, *T*15, *T*16, *T*18, *T*19, *T*20, *T*21

**Table 7 sensors-22-04442-t007:** Selected texture features on images with magnification ×100 LM.

Method for Selection	Selected Features
FSNCA	*T*5, *T*6, *T*13, *T*15, *T*19
RelieFf	*T*5, *T*6, *T*12, *T*13, *T*16, *T*19, *T*22
SFCPP	*T*2, *T*3, *T*9, *T*10, *T*11, *T*13, *T*14, *T*15, *T*16, *T*17, *T*19, *T*21, *T*22

**Table 8 sensors-22-04442-t008:** Results of classification with a naïve Bayesian classifier.

Kernel Function	Magnification		×1000 LM			×500 LM			×100 LM		
		Error	*e_i_*, %	*g_i_*, %	*e*_0_, %	*e_i_*, %	*g_i_*, %	*e*_0_, %	*e_i_*, %	*g_i_*, %	*e*_0_, %
	Selection Method										
Simple	FSNCA		0%	0%	0%	0%	0%	0%	0%	0%	0%
	RelieFf		0%	0%	0%	0%	0%	0%	0%	0%	0%
	SFCPP		0%	0%	0%	0%	0%	0%	0%	0%	0%
Poly	FSNCA		0%	0%	0%	0%	0%	0%	0%	0%	0%
	RelieFf		0%	0%	0%	0%	0%	0%	0%	0%	0%
	SFCPP		0%	0%	0%	0%	0%	0%	0%	0%	0%
Gaussian	FSNCA		13%	41%	38%	20%	41%	37%	23%	44%	42%
	RelieFf		20%	41%	40%	17%	19%	27%	30%	32%	37%
	SFCPP		20%	43%	42%	53%	33%	38%	33%	49%	52%

**Table 9 sensors-22-04442-t009:** Results of classification with discriminant analysis.

Selection Method	Separation Function	Kernel Function		Simple			Poly		
			Magnification	×1000 LM	×500 LM	×100 LM	×1000 LM	×500 LM	×100 LM
		Error							
FSNCA	Linear	*e_i_*, *%*		0%	0%	0%	0%	0%	0%
		*g_i_*, *%*		0%	0%	0%	0%	0%	0%
		*e*_0_, *%*		18%	18%	18%	18%	18%	18%
	Quadratic	*e_i_*, *%*		0%	0%	0%	0%	0%	0%
		*g_i_*, *%*		0%	0%	0%	0%	0%	0%
		*e*_0_, *%*		18%	18%	18%	18%	18%	18%
RelieFf	Linear	*e_i_*, *%*		0%	0%	0%	0%	8%	0%
		*g_i_*, *%*		0%	0%	0%	0%	0%	0%
		*e*_0_, *%*		18%	18%	18%	18%	24%	18%
	Quadratic	*e_i_*, *%*		0%	0%	0%	0%	8%	0%
		*g_i_*, *%*		0%	0%	0%	0%	0%	0%
		*e*_0_, *%*		18%	18%	18%	18%	24%	18%
SFCPP	Linear	*e_i_*, *%*		0%	0%	0%	0%	0%	0%
		*g_i_*, *%*		0%	0%	8%	0%	0%	0%
		*e*_0_, *%*		18%	18%	19%	18%	18%	18%
	Quadratic	*e_i_*, *%*		0%	0%	0%	0%	0%	0%
		*g_i_*, *%*		0%	0%	8%	0%	0%	0%
		*e*_0_, *%*		18%	18%	19%	18%	18%	18%

**Table 10 sensors-22-04442-t010:** Results of classification with support vector machines method.

Selection Method	Separation Function	Kernel Function		Simple			Poly		
			Magnification	×1000 LM	×500 LM	×100 LM	×1000 LM	×500 LM	×100 LM
		Error							
FSNCA	Linear	*e_i_*, %		0%	0%	0%	0%	0%	0%
		*g_i_*, %		0%	0%	0%	6%	0%	0%
		*e*_0_, %		0%	0%	0%	25%	0%	0%
	Quadratic	*e_i_*, %		0%	0%	0%	0%	0%	0%
		*g_i_*, %		0%	0%	0%	0%	0%	0%
		*e*_0_, %		0%	0%	0%	0%	0%	0%
	RBF	*e_i_*, %		0%	0%	0%	0%	0%	0%
		*g_i_*, %		0%	0%	0%	0%	0%	0%
		*e*_0_, %		0%	0%	0%	0%	0%	0%
RelieFf	Linear	*e_i_*, %		0%	0%	0%	0%	0%	0%
		*g_i_*, %		0%	29%	0%	12%	0%	0%
		*e*_0_, %		0%	49%	0%	34%	0%	0%
	Quadratic	*e_i_*, %		0%	0%	0%	0%	0%	0%
		*g_i_*, %		0%	0%	0%	0%	0%	0%
		*e*_0_, %		0%	0%	0%	0%	0%	0%
	RBF	*e_i_*, %		0%	0%	0%	0%	0%	0%
		*g_i_*, %		0%	0%	0%	6%	0%	0%
		*e*_0_, %		0%	0%	0%	0%	0%	0%
SFCPP	Linear	*e_i_*, %		0%	0%	0%	0%	0%	0%
		*g_i_*, %		0%	0%	0%	0%	0%	0%
		*e*_0_, %		0%	0%	0%	0%	0%	0%
	Quadratic	*e_i_*, %		0%	0%	0%	0%	0%	0%
		*g_i_*, %		0%	0%	0%	0%	0%	0%
		*e*_0_, %		0%	0%	0%	0%	0%	0%
	RBF	*e_i_*, %		0%	0%	0%	0%	0%	0%
		*g_i_*, %		0%	0%	0%	0%	0%	0%
		*e*_0_, %		0%	0%	0%	0%	0%	0%

## Data Availability

The data presented in this study may be obtained on request from the corresponding author.

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
