# Peer review of "Interdisciplinary Research to Advance Digital Imagery and Natural Compounds for Eco-Cleaning and for Preserving Textile Cultural Heritage"

_sensors, 2022, doi:10.3390/s22124442_

Round 1

Reviewer 1 Report

I do not recommend this manuscript for publication due to serious flaws. Here are my main concerns regarding the manuscript:

1- The abstract is unclear and full of full of grammatical errors. It should be rewritten.

2- The problem statement, and the motivation of the work should be presented in the introduction section.

3- The authors should use appropriate citation. Writing 8 lines in introduction section and citing like [12-29] is not the best way to refer previous works. They should cite each new statement with a proper reference.

4- Refs [6,7] are not cited in the manuscript.

5- The material and methods section should be rewritten. I strongly recommend to use an English editing service.

6- All the features presented in table 1 are global image features. In order to better analysis of image texture, local features should be utilized.

Author Response

Dear reviewer,

first of all we would like to thank you for your valuable suggestions and comments. For us, these represent opportunities for material improvement and we have tried to closely implement all your indications.

Reviewer 2 Report

  • Regarding Title: It is too generic. Kindly reframe the title and make it more specific which can explain the exact work of the manuscript
  • Kindly update the abstract to highlight the novelty of work and also highlight the performance indicator to be achieved through this study and how is this study is useful to the industry as well as researchers
  • Kindly enhance the State-of-the-art. The State-of-the-art is very limited. In this regard, kindly include at least one summary table as per the application, which will include the following information at least: Reference number, year of publication, domain, main contribution, enhancement requirement/disadvantage, similarity with proposed work. Then list out all possible research gap findings. Then mention at least one research approach, which will overcome at least one research gap finding in this paper.
  • The implementation of the proposed approach is not clear. Kindly explain in detail in a step-wise-step manner with mathematical modelling and examples.
  • Kindly include a proposed approach and include a diagram which can explain whole work of this paper in step-wise-step manner
  • Need to include some market analysis related to this work

Author Response

(The authors gave the same response as above.)

Round 2

Reviewer 1 Report

The authors made some modifications in this version. I do not observe any technical improvement in the manuscript. Here are some major issues regarding this work:

- From machine learning point of view, this work is a binary classification task. First of all, the number of samples are handful. Does this amount of data is enough to be even partitioned into training and test sets??

- There is no consistency between the results of figure 7 and figure 8 for washed and unwashed samples.

- The title of table 3 is irrelevant. Also, the formulation of features are ambiguous and incorrect for some cases. Could the authors please clarify that how, for instance, T2 represents image contrast?

- I strongly recommend that the authors follow the procedure of image classification using feature extraction in the literature. The way presented in this work, is not even close to machine learning based project.

- some references are added to bibliography section but the order and numbering in the body has not been changed

Author Response

(The authors gave the same response as above.)

Reviewer 2 Report

Authors have updated the required comments. I don't have any comment now.

Author Response

Dear reviewer,

thank you for your attention and your constructive suggestions.

Best regards

Round 3

Author Response

(The authors gave the same response as above.)
